# Quartz-Enhanced Photothermal Spectroscopy-Based Methane Detection in an Anti-Resonant Hollow-Core Fiber

**DOI:** 10.3390/s22155504

**Published:** 2022-07-23

**Authors:** Piotr Bojęś, Piotr Pokryszka, Piotr Jaworski, Fei Yu, Dakun Wu, Karol Krzempek

**Affiliations:** 1Faculty of Electronics, Photonics and Microsystem, Wroclaw University of Science and Technology, 50-370 Wroclaw, Poland; piotr.pokryszka@pwr.edu.pl (P.P.); piotr.jaworski@pwr.edu.pl (P.J.); karol.krzempek@pwr.edu.pl (K.K.); 2Hangzhou Institute for Advanced Study, University of Chinese Academy of Sciences, Hangzhou 310024, China; yufei@siom.ac.cn (F.Y.); wudakun@ucas.ac.cn (D.W.); 3Key Laboratory of Materials for High Power Laser, Shanghai Institute of Optics and Fine Mechanics, Chinese Academy of Sciences, Shanghai 201800, China

**Keywords:** quartz-enhanced photothermal spectroscopy (QEPTS), anti-resonant hollow-core fiber (ARHCF), quartz tuning fork

## Abstract

In this paper, the combination of using an anti-resonant hollow-core fiber (ARHCF), working as a gas absorption cell, and an inexpensive, commercially available watch quartz tuning fork (QTF), acting as a detector in the quartz-enhanced photothermal spectroscopy (QEPTS) sensor configuration is demonstrated. The proof-of-concept experiment involved the detection of methane (CH_4_) at 1651 nm (6057 cm^−1^). The advantage of the high QTF Q-factor combined with a specially designed low-noise amplifier and additional wavelength modulation spectroscopy with the second harmonic (2f-WMS) method of signal analysis, resulted in achieving a normalized noise-equivalent absorption (NNEA) at the level of 1.34 × 10^−10^ and 2.04 × 10^−11^ W cm^−1^ Hz^−1/2^ for 1 and 100 s of integration time, respectively. Results obtained in that relatively non-complex sensor setup show great potential for further development of cost-optimized and miniaturized gas detectors, taking advantage of the combination of ARHCF-based absorption cells and QTF-aided spectroscopic signal retrieval methods.

## 1. Introduction

Laser spectroscopy of gases has been a strongly developed group of measurement techniques for many years [1,2,3], allowing the measurement of the composition of various gas mixtures. Due to the strong need to control the presence of, e.g., greenhouse gases [4,5,6], it has been successively developed both by introducing new techniques and by improving those already commonly used. Simple sensors use single-pass configurations, that are highly limited by the footprint of the device or the detector, limiting the maximum achievable open-path distance. High sensitivity is obtainable in miniature sensors, however, at the cost of their complexity [7,8,9,10]. One of the classical ways to achieve lower detection limits of an absorption-based laser gas sensor is to extend the optical path. According to Lambert-Beer’s law [11], the absorption of a medium is proportional to the interaction distance between the laser beam and the gas particles; thus, the extension of the optical path leads to higher absorption of the gas analyte, which, in the end, results in improvement of the detection limit of the device. The classic approach of increasing the optical path in a spectroscopic system is to use a multi-pass cell (MPC). Unfortunately, MPCs have disadvantages: they are usually heavy and unwieldy, require relatively large volumes of the tested gas sample, and may introduce additional optical noise [12]. For this reason, alternative solutions are being extensively explored.

One of them is based on the use of hollow-core fibers (HCFs), which simultaneously guide the laser beam performing the measurement and can be filled with the gas sample under test. Several configurations of such unique sensors have been developed and acquired significant attention throughout the laser spectroscopy community. For this purpose, the photonic bandgap fibers (PBG) have been widely used, although two main problems were observed. The first one is the limited possibility of transmittance in the mid-infrared (mid-IR) light spectrum [13], which is highly desirable in gas spectroscopy [14,15]. The other one results from the multimode nature of light transmission [16], which can be problematic in most of the spectroscopic techniques due to the presence of intermodal interference. A breakthrough was achieved with the development of the anti-resonant hollow-core fibers (ARHCFs), in which transmission is based on a model known as ARROW (anti-resonant reflecting optical waveguide) [17]. It is a special type of hollow-core fiber, characterized by a wide achievable transmission spectrum (even above 5 μm [10,18]) with quasi single-mode transmission [19] and large air-cores, which simplify the gas exchange process. The above-mentioned advantages attracted researchers worldwide, who experimentally confirmed the usefulness of the ARHCFs in the development of highly sensitive gas detectors based on various configurations and spectroscopic techniques [20,21,22]. The results show that, with proper design, sensors can achieve excellent sensitivity, mitigating the use of troublesome MPCs [22]. However, all the above-mentioned gas sensors require a low-noise detector to convert the spectroscopic signal encoded to the optical wave to the electrical signal interpretable by laboratory apparatus, e.g., lock-in amplifiers. This is usually accomplished by using semiconductor-based photodetectors. These devices are based on the intrinsic photoelectric phenomenon, whereby an electric signal is generated by a photon incident on the surface of a semiconductor junction. The major limitation of these types of detectors is their operating wavelength range, and in some cases the radiofrequency bandwidth. The material of which the photodiode is made determines the range of wavelengths that can generate an electrical signal [23,24]. Moreover, mid-IR detectors are still expensive, require low operating temperatures to maintain their low-noise characteristic, and are manufactured from hazardous materials [25,26]. An alternative solution is to use the so-called quartz tuning fork (QTF) acting as a traditional photodetector.

A QTF is a type of resonator made of a piezoelectric silicon dioxide crystal [27]. One of the most popular applications of the QTF is as a resonator in oscillators, such as those used in watches, where the QTF, through the use of the inverse piezoelectric effect [28], is implemented as a frequency-determining element. If the QTF is subjected to an external force acting on its prongs, an electric charge due to the piezoelectric phenomenon [28] will be generated. The strongest excitation and electrical response will be observed only when the excitation force will be periodic and have a frequency consistent with the resonant frequency of the resonator [29]. Those effects have been adapted in unique configurations of laser-based gas detectors. Numerous papers have been reported on the quartz-enhanced photoacoustic spectroscopy (QEPAS) technique [30], in which the QTF is used to detect an acoustic wave generated between its prongs as a result of laser beam absorption in the gas sample. QEPAS-based sensors have been shown to deliver very high sensitivity and selectivity in the detection of gas molecules in the near- and mid-IR spectral bands [31,32]. A variation of this technique is called quartz-enhanced photothermal spectroscopy (QEPTS) [33] or light-induced thermoelastic spectroscopy (LITES) [34], where the QTF is used as a high Q-factor detector, alternatively to a traditional semiconductor-based photodetector. In the QEPTS sensing technique, the excitation of the QTF resonator is caused by locally heating its surface with a laser beam previously transmitted through the gas sample under test (the wavelength of the laser is matched to the absorption of the gas particles). Due to the thermoelastic expansion of the quartz, the QTF is temporarily deformed in the luminescent region. In such sensors, the laser beam is modulated at the resonant frequency of the QTF, taking advantage of high Q-factor noise rejection. Due to the piezoelectric phenomenon, an electrical signal is generated at the electrodes of the QTF, which is proportional to the amplitude of the laser beam inducing the thermoelastic effect. In a sensor setup, prior to illumination of the prong of the QTF, the beam is firstly transmitted through a gas sample, and as a result the information about the analyte concentration can be directly encoded to the QTF-generated electrical signal interpretable by, e.g., a lock-in amplifier, accompanied by, e.g., second harmonic-based signal readout. Due to the high-frequency selectivity (high Q-factor), the broad absorption spectrum of quartz (including absorption at shorter wavelengths) [35], and the low cost and wide availability, the QEPTS detection technique is an interesting alternative for traditional gas sensors, even when compared to QEPAS in terms of sensitivity and complexity [30].

In this paper, we propose a unique laser-based gas sensor that benefits both from the QEPTS detection technique and the implementation of an ARHCF as an absorption cell. As a proof-of-concept, detection of methane (CH_4_) at 1651 nm (6057 cm^−1^) using a wavelength modulation spectroscopy method (WMS) [36] at the second harmonic (2f) was demonstrated. The sensor reached a noise-equivalent absorption (NEA) of 8.23 × 10^−8^ cm^−1^ for a 100 s integration time, while maintaining a non-complex design.

## 2. Materials and Methods

### 2.1. Materials

The fiber used as an absorption cell is a nodeless-type ARHCF, whose cross-section scanning electron micrograph (SEM) is depicted in Figure 1a. The fiber has an air-core with a diameter of 84 μm and a length of 0.95 m. The hollow-core region is made up of seven nonadjacent capillaries, each with ~1 μm-thick walls. According to the ARROW guidance mechanism, the fiber has two low-loss transmission bands located in the near- and mid-IR spectral regions. In this work, we focus on the use of the first-order transmission band spanning from ~ 1150 to 1700 nm, as presented in Figure 1b. The fiber is characterized by low loss, not exceeding 2 dB/m within the entire guidance window, which is sufficient for the proposed gas sensing approach. More information about this particular fiber can be found in [37,38].

The spectroscopic signal was retrieved using the QEPTS technique with a simple watch QTF, shown in Figure 1c, with a resonance frequency of ~32.768 kHz. Before being implemented into the sensor setup, the resonator was stripped of its cylindrical aluminum shell, allowing us to expose the prongs. As a result, the resonance frequency of the fork changed slightly in comparison to its nominal value (see Section 2.3 for further details). Furthermore, since the QTF is coated with a thin layer of silver, which is characterized by strong light reflection properties, it was necessary to determine the optimal area within which it can be illuminated by the laser radiation and, consequently, efficiently heated. This is necessary to obtain high signal amplitude at the electrodes and, subsequently, a greater detection capability of the sensor. The area where the laser beam was focused is marked as a red dot in Figure 1c.

The electric signal gathered from a bare QTF is miniscule; thus, an appropriate amplifier is required to achieve satisfactory performance of the sensor. The self-made QTF amplifier, for which a simple block scheme is presented in Figure 2, consists of a buffering circuit, an active filter with simultaneous amplification of the signal, an amplifier circuit with the possibility of modifying the gain, and a passive filter. The construction of the signal amplifier module for the QTF is described below. A small-amplitude signal coming from the excited QTF is passed to the AD8244 buffer circuit. The unit-gain operational amplifier circuit used in the experiment has an input impedance of 10 TΩ and low noise levels. The signal is then passed to an active bandpass filter with a gain of 60 *v*/*v*. A two-stage Butterworth-type bandpass filter with a center frequency of 32.744 kHz and a bandwidth of 2 kHz is used. The chosen bandwidth considers the technological spread of the resonant frequency of the QTF, and the passive elements used for the active filter. The Butterworth-type filter was used because of its flat amplitude characteristics in the passband. An AD8676 operational amplifier was used for the active filter. The LT6372 instrumentation amplifier circuit that was used after the active filter is designed to allow modifying the signal gain during the experiments, without soldering. The output signal is then delivered to a passive filter with a cut-off frequency of 46 kHz.

### 2.2. Experimental Setup

The experimental setup of the developed CH_4_ sensor is depicted in Figure 3.

A tunable distributed feedback diode laser (EP1651-0-DM-B01-FA, Eblana Photonics Ltd., Dublin, Ireland) with emission centered at 1651 nm was used as a CH_4_ molecules excitation source. The laser parameters, that is, the injection current and the operation temperature, were controlled by a commercial laser driver (CLD1015, Thorlabs Inc., Newton, NJ, USA), which allows tuning of both the aforementioned parameters. The wavelength of the laser-emitted radiation was tuned to the center of the selected CH_4_ transition at 1651 nm. The laser light was coupled to the fiber with a fiber collimator (output beam diameter = 12 mm) and an N-BK7 lens (f = 200 mm) to match the acceptance angle of the ARHCF. The optical power measured at the end of the fiber filled with N_2_ was at the level of 2.4 mW. The input end-face of the ARHCF was glued to a special, self-designed gas cell, with an N-BK7 optical wedge used as the input aperture. The gas cell enabled filling the fiber via the pressure-induced flow method [39]. An additional purge port (PP in Figure 3) provides the possibility of purging the gas cell before each measurement. The length of the ARHCF used in the experiment was 0.95 m. Due to the purge port, the gas exchange time in the fiber was only 10 s for the 25 Torr overpressure, as indicated in Figure 4. The volumetric flow rate through the air-core (air-core area = 0.0193 mm^2^) was at the level of 0.0017 mm^3^/s. The other end of the ARHCF was secured on a 5-axis micro-positioning stage (PY005, Thorlabs), which enabled outcoupling the laser beam directly to the QTF placed at a distance of ~0.2 mm, perpendicular to the fiber’s end-facet, without using additional optics. The QTF was installed on a self-made amplifier (described earlier).

To take advantage of the high Q-factor, the laser beam has to induce the thermoelastic effect with a frequency matching the QTF resonant frequency, f_r_ = 32.744 kHz. In this experiment, the laser beam was modulated with a sinewave signal at half of the f_r_/2 = f_mod_ = 16.372 kHz to implement the WMS detection technique. The signal was generated using an arbitrary waveform generator (AFG 3102, Tektronix, Inc., Beaverton, OR, USA) along with a slow saw-tooth ramp signal (f_ramp_ = 0.2 Hz) used to scan the laser wavelength to retrieve the full absorption profile of the CH_4_ transition (the voltage signal was coupled to the modulation input of the laser driver). When the laser beam is absorbed in the vicinity of the CH_4_ transition, harmonics of the f_mod_ frequency are generated, with the 2f harmonic (2 × f_mod_) having its maximum at the center of the gas absorption line [36,40], and thus taking advantage of the QTF high Q-factor at this frequency, which significantly improves the detected signal characteristic. A lock-in amplifier (SR830, Stanford Research System, Sunnyvale, CA, USA) was used to extract the signal at the 2 × f_mod_ frequency and to record it on a digital oscilloscope (model RTB2004, Rohde & Schwarz GmbH & Co KG, Munich, Germany).

The surface of the QTF was not modified, with the prongs almost completely coated with a silver contact layer. Only a few uncoated spots remained, where the laser beam could be directed without the risk of back-reflections and additional noise caused by scattering at the glass structure of the fiber. Due to the fact that the strength of thermoelastic-induced oscillations depends on the position of the incident beam on the QTF surface, it was crucial to place the QTF properly, relative to the output of the ARHFC [33]. The optimal position of QTF was experimentally chosen by observation of the 2f signal amplitude registered for a 2000 ppmv (parts per million by volume) mixture of CH_4_ and nitrogen (N_2_), with special attention paid to both the signal amplitude and the signal oscillations that occur with suboptimal QTF settings (optical fringe caused by back-reflections from the silver layer on the QTF). Changing the illumination position can to some extent affect the level of thermal noise generated by the quartz resonator [41,42]. In the case of the presented system, the maximum signal-to-noise ratio occurred at a point of maximum 2f signal amplitude. As can be seen in Figure 1c, the heating spot that was chosen in our experiment matches the one reported in [33], as one of the acceptable heating points.

### 2.3. Sensor Parameters’ Optimization

In the QEPTS technique, one of the main parameters that must be properly defined is the f_r_ of the QTF. Due to the high Q-factor of a typical crystal resonator, the piezoelectric signal generation efficiency will be highly negatively affected by off-tuning of the spectroscopic signal from this frequency. As described before, as a consequence of the de-canning of the QTF, the f_r_ deviated from the value provided in the device datasheet, and thus it was necessary to experimentally determine the optimal modulation frequency.

Due to the implementation of the 2f-WMS technique, we decided to optimize the modulation frequency for the target measuring frequency, which was equal to half of the resonance frequency of the QTF (f_mod_ = f_r_/2). The sweep of the sinewave modulation frequency was performed between 16.25 and 16.50 kHz, with simultaneous observation of the LIA 2f signal amplitude. For measurements, the fiber was filled with a mixture of 2000 ppmv CH_4_. As can be seen in Figure 5a, the Q-factor obtained for the resonator was 7818 and the optimum f_mod_ for the QTF was found to be 16.372 kHz. The Q-factor is given by the following formula [43]:(1)$$Q=\frac{f_{\mathrm{mod}}}{\Delta f}$$
where f_mod_ is the optimal modulation frequency for the 2f-WMS measuring technique, equal to half of the resonance frequency, and Δf is the width of the response characteristic at its half-height.

Another parameter that was crucial for the optimization of the presented setup is the wavelength modulation depth, which was controllable by changing the amplitude of the sinewave voltage signal delivered from the arbitrary function generator to the laser driver. In the WMS technique, the proper adjustment of this parameter determines the maximum achievable signal level, which leads to better detectability of the designed system (due to the higher SNR). During this measurement, the time constant of the LIA was set at 1 ms and the fiber was filled with 2000 ppmv of CH_4_, and the laser wavelength was tuned to the center of the absorption line. As can be seen in Figure 5b, the wavelength modulation depth was swept from 2.5 to 25.0 GHz. The optimal modulation depth that was determined from this measurement was 10.6 GHz.

In the next step, the full 2f-WMS signal was acquired using the optimal sinewave modulation parameters defined above, with the laser wavelength sweeping through the absorption line of the gas analyte. The concentration of the CH_4_ mixture inside the ARHCF’s core was 2000 ppmv. Subsequently, the ARHFC was filled with pure N_2_ at an overpressure of 25 Torr and the baseline noise of the sensor was measured for 2000 s with the time constant to 1 ms on the LIA. Other measurement parameters are listed in Figure 6. The small offsets shown in Figure 6 resulted from the effect of photo-thermoelastic energy conversion induced by absorption of the laser beam by the QTF [44]. The calculated 1σ standard deviation of the noise was 912 µV.

## 3. Results

### 3.1. Sensor Linearity

The linearity of the senor response was verified by registering the 2f signal for various concentrations of CH_4_ inside the ARHCF-based absorption cell. Gas samples were prepared using certified gas mixtures (from Air Liquide) and a commercial gas mixer (Environics 4000, Environics, Toronto, ON, Canada) in the following variations: 8000, 4000, 3000, 2000, 1000, 200, 100, and 30 ppmv. Before each measurement, the ARHCF was purged with pure N_2_ from the previous mixture. The time constant on the LIA was set to 1 ms for all measurements. Figure 7a shows the results obtained for each gas concentration, and Figure 7b shows the maximum registered amplitudes plotted as a function of concentration. During measurements, CH_4_ pressure was set at 25 Torr of overpressure. With a linear approximation of the obtained data points, R^2^ was calculated at a value of 0.997, which confirms the good linearity of the proposed system over a broad measurement range [45,46].

### 3.2. Long-Term Sensor Performance

The last aspect that was measured is the stability of the system and the resulting detection limit. For this purpose, the amplitude of the 2f noise signal of the system was continuously measured for 20 min, during which the ARHCF was filled with pure N_2_. The wavelength modulation parameters were set to the optimal values determined for the selected CH_4_ absorption line (sinusoidal modulation with a depth of 10.6 GHz and a frequency of 16.372 kHz), and the laser wavelength was tuned to the center of the absorption line (no active stabilization was used). The time constant of the LIA was set to 1 ms, with an 18 dB/oct filter setting. Based on the measurements, the Allan–Werle deviation was plotted, which is presented in Figure 8b as a black trace. The Allan–Werle deviation is a commonly used measure of sensor stability in various spectroscopic techniques, which enables determining the maximum achievable sample integration time for the proposed system [47]. This, therefore, provides an estimate of the detection limit for a selected integration time of the sensor. The analysis is usually performed by integrating samples from the noise signal amplitude acquired during a long-time measurement, e.g., over several tens of minutes, while the sensor’s operating parameters are set as the optimal values for the detection of the target gas. During this measurement, the gas cell is usually filled with N_2_. The values on the y-axis are presented in ppmv for clarity reasons (calculated based on the maximum 2f signal obtained for 2000 ppmv of CH_4_). The minimum detection limit (MDL) of the proposed system is at the level of 1.44 and 220 ppbv (parts per billion by volume) for 1 and 100 s of integration time, respectively. Based on the measurements, the noise-equivalent absorption (NEA) obtained was at a level of 5.39 × 10^−7^ and 8.23 × 10^−8^ cm^−1^ for the 1 and 100 s integration times, respectively. The normalized noise-equivalent absorption (NNEA) [45] was at the level of 1.34 × 10^−10^ and 2.04 × 10^−11^ W cm^−1^ Hz^−1/2^ for 1 and 100 s, respectively. The noise of the QTF and the custom amplifier was also investigated. This was realized by turning off the laser and registering the 2f amplitude, as in the previous experiment. The results are plotted in Figure 8b in red and blue, with and without N_2_ flow during measurement of the final result. Based on these results, we can conclude that the proposed sensor outperformed previously published similar sensors in terms of complexity and is at a similar level or better in terms of the obtained NNEA [33,34,45,46,48].

## 4. Discussion

The results presented in this work show great potential in the combination of QTF-based measurement techniques with ARHCF absorption cells. The detection capability of the developed system (NNEA of 1.34 × 10^−10^ and 2.04 × 10^−11^ W cm^−1^ Hz^−1/2^ for 1 and 100 s of integration time, respectively) beats the performance of the more complex QEPTS systems, as presented in Table 1.

Due to the simplicity of the designed system, we were able to reduce the optical noise, and we believe that there is room to simplify the design of the sensor even further. As it has been reported in [51], it is possible to splice ARHCF with a solid-core fiber with good coupling efficiency, and hence acceptable transmission loss. Moreover, as has been presented in [52], one of the options to effectively fill HCF with gas is to use additional fabricated micro-holes to improve diffusive exchange of gases. With that, there is a space to design a sensor without any additional bulk optics components, with pure diffusive gas exchange, which could significantly improve the compactness and simplicity of the designed sensor. The filling time of 10 s for the developed system indicates great potential in implementing such a sensor outside the laboratory, where near real-time response times are desirable. Furthermore, there are several aspects that can be improved in the next design of the ARHCF-QEPTS sensing system. The fiber was not perfectly optimized for the selected wavelength (it operates at the edge of the transmission window), and its length was relatively short. Therefore, there is a great potential to improve the system by using an ARHCF designed directly for the wavelength used, which should reduce the optical noise in the sensor. When a longer fiber is used, it may be possible to obtain a higher absorption signal for the same concentration, which ultimately increases the detection limit of the sensor. Additionally, in our system, the QTF was not isolated from the external environment, which could also affect the noise contribution of the setup and its overall stability. Furthermore, the signal level can be further increased by using multiple QTFs as thermoelastic detectors.

## 5. Conclusions

In summary, in our work we demonstrated a combination of two advanced laser gas spectroscopy techniques, i.e., the application of an ARHCF-based absorption cell and implementation of a QTF detector operating on the principle of the QEPTS technique. Thus, we were able to eliminate both the bulk optics MPC and the expensive, narrowband infrared detector from the system. The use of a QTF resonator instead of a detector makes it feasible to obtain a relatively inexpensive (the cost of a single watch QTF is at the level of a few cents) and broadband detector, allowing absorption measurements both in the near- and mid-infrared regions. The obtained results show that the use of an appropriately designed amplifier allows obtaining NNEA at the level of 2.04 × 10^−11^ W cm^−1^ Hz^−1/2^, which is comparable or better to other proposed and usually more advanced QTF-based measurement systems [31,45]. Additionally, it is noticeable that the use of ARHCF, although not optimized for the considered wavelength range, did not cause a significant decrease in the detection capabilities of the system. This suggests further possibilities of the development of the system by using both a fiber with better transmission characteristics and a better isolated and more accurate detection module. The next step will be to expand the system to include the ability to simultaneously detect more than one gas targeting the absorption lines in the mid- and near-infrared spectral bands, which should show even more clearly how much potential the combination of QEPTS-based detectors with ARHCF absorption cells offers.

## Figures and Tables

**Figure 1 sensors-22-05504-f001:**
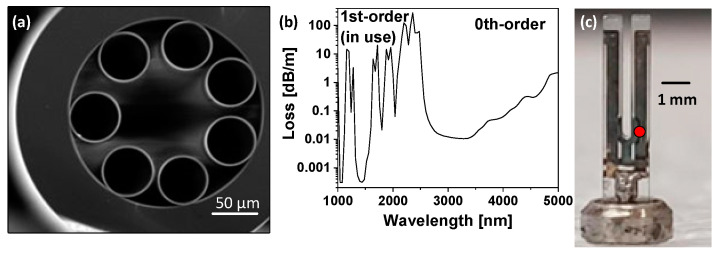
(**a**) SEM image of the used ARHCF. (**b**) ARHCF’s loss characteristic. (**c**) Photograph of the implemented QTF. The red dot marks the optimal heating area.

**Figure 2 sensors-22-05504-f002:**
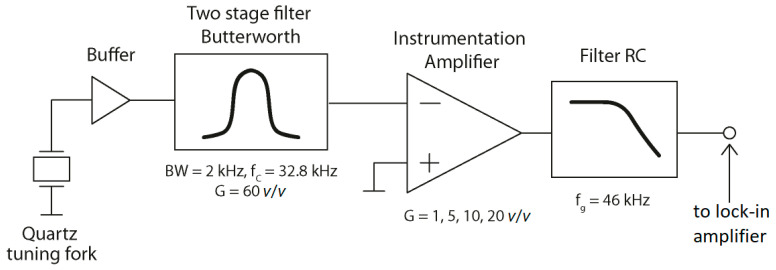
Block diagram of the amplifier used in the experiment. BW—bandwidth; f_c_—central frequency; G—gain; f_g_—cut-off frequency of the RC filter.

**Figure 3 sensors-22-05504-f003:**
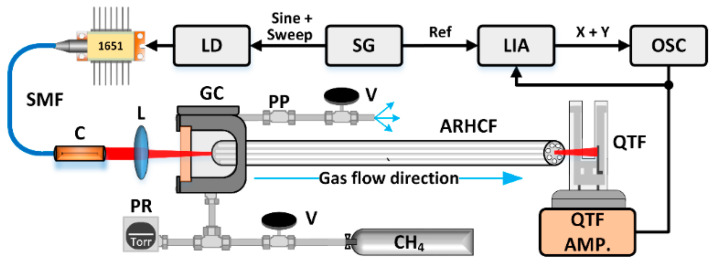
Schematic of the sensor setup. 1651—fiber-pigtailed distributed feedback laser diode; LD—laser driver; SG—signal generator; LIA—lock-in amplifier; OSC—oscilloscope; C—collimator; L—lens; GC—gas cell; PP—purge port; V—valve; PR—pressure readout; QTF AMP.—QTF amplifier.

**Figure 4 sensors-22-05504-f004:**
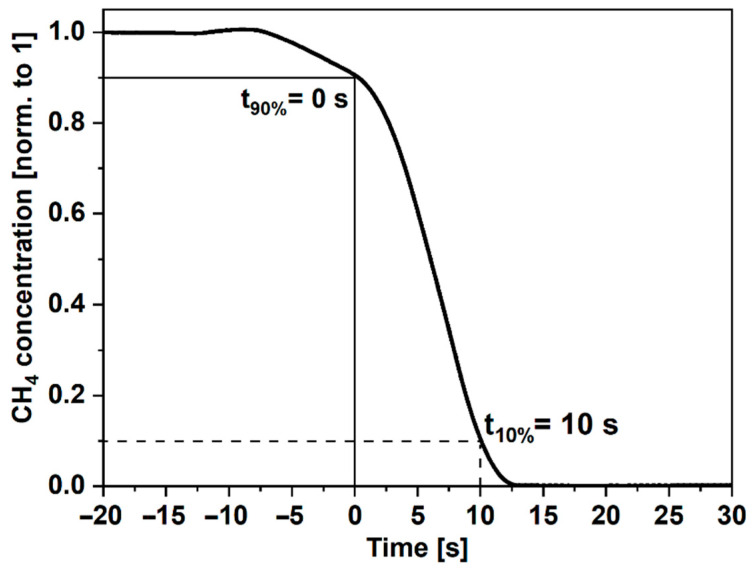
Gas exchange time measured for N_2_ pressured through the fiber with 25 Torr overpressure.

**Figure 5 sensors-22-05504-f005:**
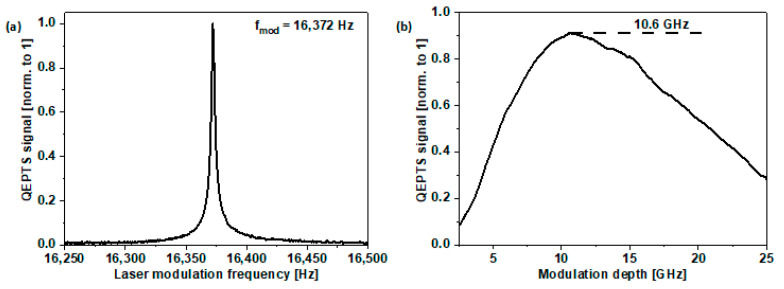
(**a**) Characterization of the QTF frequency responsivity. (**b**) QEPTS signal amplitude registered for different values of sinewave modulation depth. Measurements were performed with ARHCF filled with 2000 ppmv of CH_4_ and f_mod_ = 16.372 kHz.

**Figure 6 sensors-22-05504-f006:**
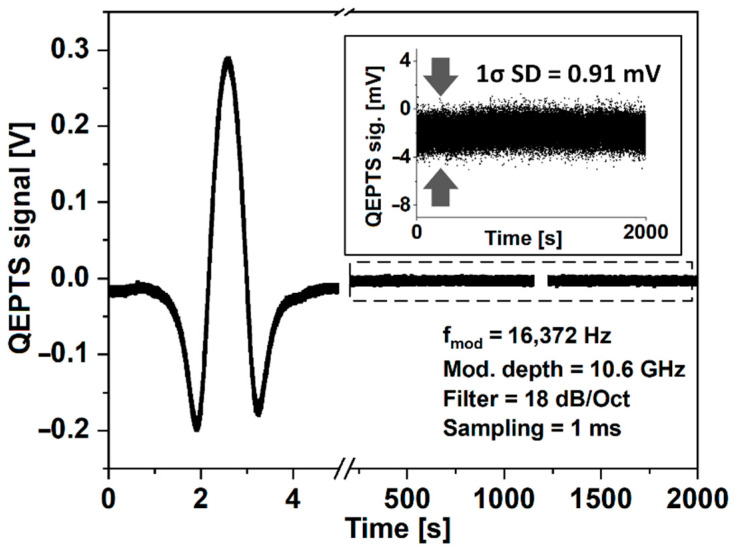
The QEPTS signal recorded for 2000 ppmv of CH_4_ in the ARHCF acquired with optimal modulation parameters was followed by a 2000 s noise amplitude measurement registered with N_2_ in the fiber. The inset shows a close-up of the registered noise amplitude for clarity.

**Figure 7 sensors-22-05504-f007:**
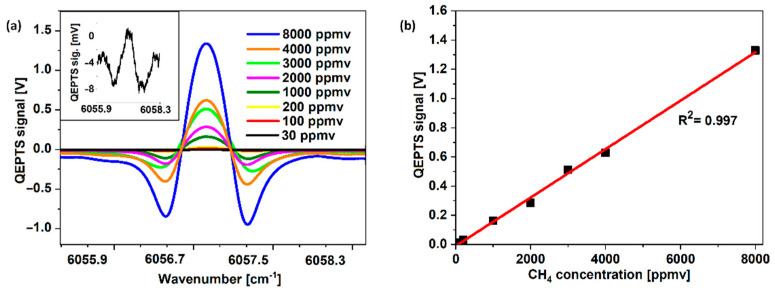
(**a**) Full 2f-WMS scans across the CH_4_ absorption line registered for concentrations listed in the graph (pressure of gas on the inlet of fiber during registration: 25 Torr over atmospheric pressure). (**b**) Maximum 2f-WMS amplitude plotted as a function of CH_4_ concentration showing the linearity of the response of the proposed sensor configuration. The obtained R^2^ value is listed on the graph.

**Figure 8 sensors-22-05504-f008:**
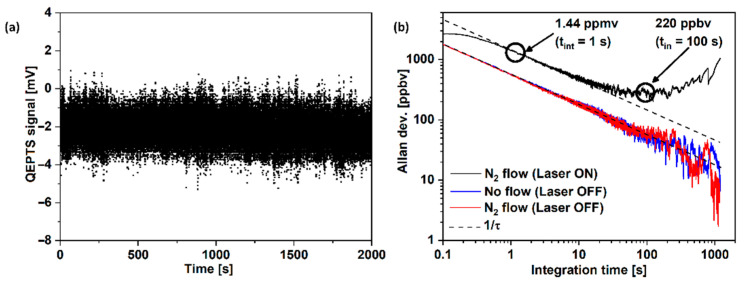
(**a**) Noise amplitude registered with N_2_ flowing through the sensor. (**b**) Allan deviation plot calculated from noise registered with N_2_ flowing through the sensor (black trace), with N_2_ flowing through the sensor but with the laser turned off (red trace), and without the flow of N_2_ (blue trace).

**Table 1 sensors-22-05504-t001:** Comparison of the performance of gas sensors utilizing QEPTS.

Absorp. Cell	Gas	Integ. Time	MDL	NNEA (W cm^−1^ Hz^−1/2^)	Ref.
MPC 10.1 m	CO	60 ms	470 ppbv	2.0 × 10^−7^	[34]
Single-pass 0.2 m	C_2_H_2_	200 s	190 ppbv	−	[46]
MPC 40 m	CO	200 s	9 ppmv	1.15 × 10^−7^	[49]
MPC 10.1 m	CH_4_	100 s	400 ppbv	−	[50]
Single-pass 2 m	H_2_O	−	12 ppmv	8.4 × 10^−7^	[48]
ARHCF (0.95 m)	CH_4_	100 s	220 ppbv	2.04 × 10^−11^	This work

## Data Availability

Data underlying the results presented in this paper are not publicly available at this time but may be obtained from the authors upon reasonable request.

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
