# Peer review of "Quartz-Enhanced Photothermal Spectroscopy-Based Methane Detection in an Anti-Resonant Hollow-Core Fiber"

_sensors, 2022, doi:10.3390/s22155504_

Round 1
Reviewer 1 Report
The article, “Quartz-Enhanced Photothermal Spectroscopy based methane detection in an Antiresonant Hollow-Core Fiber” presented a novel gas sensor with quartz-enhanced photothermal spectroscopy technology and antiresonant hollow-core fiber. The experimental results are good compared with the multipass cell. However, the English expression approach of this manuscript is difficult to understand, such as the first sentence of the Section Introduction “Laser spectroscopy of gases has been for many years a strongly developed group of measurement techniques”. In addition, the below questions should be improved:
1. A antiresonant hollow-core fiber was used in this work as an absorption cell and enhanced the targeted gas absorption path. However, the results of the comparison with bare QTF were not demonstrated in the manuscript. If the improvement of signal amplitude was not conspicuous, the employ of the complex hollow-core fiber is not advocated.
2. In Fig. 1c, a red dot was marked as the optimal area to detect signal amplitude. However, the optimal area should be defined as the largest signal-noise ratio area, since a larger background noise also can be observed when the largest signal amplitude was observed.
3. It is better to add the signal amplitude comparison of with and without using the self-made QTF amplifier.
4. Line 110 “The fiber has an air core with a diameter of 84 μm and a length of 0.95 m”. Why uses these parameters? The optimization of the fiber parameters is interesting to reads.
5. The English writing should be improved. Many sentences read like spoken language.
Reviewer 2 Report
This manuscript by Piotr et al provides a proof of concept for the detection of methane gas using a unique laser-based gas sensor. The introductory material and the analysis of the sensing performance is well-presented. As such, this reviewer recommends consideration for publication, after minor corrections:
1. figure 1 could be enlarged so that the actual architecture of the device is visible.
Reviewer 3 Report
The authors propose an original method for measuring the concentration of gases, in this case methan. It consists in the use of two main elements, namely, hollow core negative curvature fibers and quartz tuning fork which is used as a trditional photodetector. The results of the article really demonstrate the promise of using such a relatively simple gas sensor system. The authors conducted some very convincing experiments. I believe that the article can be published in Sensors, but I have several important comments that I would like to receive answers in the article:
1. The authors write: "It is a special type of hollow-core fiber, characterized by a wide achievable transmission spectrum (even above 5 μm [10]) with quasi single-mode transmission [18]..."
but it is known that the first demonstration of light transmission in very high loss region up to wavelength of 8 μm was published in Optics Express, 21, 9514 (2013).
2. It would be very good both for the presentation of experiments in the paper and for the general understanding of people who are not familiar with this field of optics to explain in the article at least briefly the physical meaning of the terms: QEPTS technique and Allan - Werle deviation.
Round 2
Reviewer 1 Report
The authors answered the questions, and most of them were convincing. However, for question 2, according to my experiment results, the background noise was very relevant to the laser beam position. Please double-check it.
